# Bacterial profile and antimicrobial susceptibility patterns of isolates from inanimate objects used by healthcare professionals at Debre Markos Comprehensive Specialized Hospital, Northwest Ethiopia

Senedu Kindie[1,2], Getachew Mengistu[1], Mulatu Kassahun[3], Abebaw Admasu[1], Tebelay Dilnessa[1] *

1 Department of Medical Laboratory Sciences, College of Medicine and Health Sciences, Debre Markos University, Debre Markos, Ethiopia, 2 Leul Alemayehu Health Center, Debre Tabor, Ethiopia, 3 Students' Clinic Burie Campus, Debre Markos University, Burie, Ethiopia

* tebelay_dilnessa@dmu.edu.et

## Abstract

### Background

Healthcare-associated infections occur in patients under medical care, which is a major public health issue in hospitals worldwide. The prevalence is two to three folds higher in developing countries compared to developed countries. Inanimate objects used by healthcare professionals such as gowns, mobile phones, and stethoscopes are highly prone to bacterial contamination from the healthcare setting. In Ethiopia, there is a paucity of published data on the bacterial profile and susceptibility patterns of an isolates.

### Objective

To determine the bacterial profile and antimicrobial susceptibility patterns from inanimate objects used by healthcare professionals and associated factors at Debre Markos Comprehensive Specialized Hospital, Northwest Ethiopia.

### Methods

A cross-sectional study was conducted at Debre Markos Comprehensive Specialized Hospital from inanimate objects from April 10, 2023 to June 30, 2023 using simple random sampling technique with lottery method. Socio-demographic data was collected using a structured questionnaire. A swab sample from healthcare professionals' gowns, mobile phones, and stethoscopes were collected and inoculated onto blood agar, chocolate and MacConkey agar. Isolates were identified and characterized by colony morphology, Gram staining and biochemical tests. Antimicrobial susceptibility testing was performed in all isolates by the disk diffusion technique on Muller Hinton agar according to CLSI, 2022

**Data Availability Statement:** All relevant data are within the manuscript and its Supporting Information files.

**Funding:** The author(s) received no specific funding for this work.

**Competing interests:** The authors have declared that no competing interests exist.

**Abbreviations:** AMR, Antimicrobial Resistance; APHI, Amhara Public Health Institute; ATCC, American Type Culture Collection; BSI, Bloodstream Infection; CLSI, Clinical and Laboratory Standards Institute; DMCSH, Debre Markos Comprehensive Specialized Hospital; HAI, Healthcare Associated Infection; HCP, Healthcare Professionals; ICU, Intensive Care Unit; MDR, Multidrug Resistant; MRSA, Methicillin Resistant *Staphylococcus aureus*.

guidelines. Data was entered into EPI-Data and analyzed using SPSS version 25. Logistic regression model was used. Variables with p-value ≤ 0.25 in bivariable logistic regression was fitted to multivariate logistic regression analysis. A p-value of ≤ 0.05 with 95%CI was considered statistically significant.

## Result

A total of 376 healthcare professionals' inanimate objects were included from 191 study participants. Overall, 60.1% (226/376) (95%CI: 55.1–65) inanimate objects were contaminated. The proportion of bacterial contamination was (65.1%; 82/126), (60.3%; 76/126) and (54.8%; 68/124) for mobile phones, gowns and stethoscopes, respectively. *Staphylococcus aureus* was the most frequent isolate accounting (22.1%; 83/376) followed by *Staphylococcus epidermidis* (17.0%; 64/376), *Escherichia coli* (8.8%; 33/376) and *Pseudomonas aeruginosa* (4.9%; 18/376). Working in gynecology/ obstetrics wards (AOR: 8.69; 95%CI: 1.09–69.41, $P = 0.041$), did not disinfect mobile phones (AOR: 2.69; 95%CI: 1.15–6.25; $P = 0.021$) and stethoscopes regularly (AOR: 3.06, 95%CI: 1.23–7.59; $P = 0.016$), carry mobile phones with patient care materials (AOR: 2.72; 95%CI: 1.18–6.29, $P = 0.019$) and not taken infection prevention training (AOR:3.91; 95%CI:1.71–8.93; $P = 0.001$) were significantly associated with bacterial contamination.Most Gram-negative bacteria were resistant to ampicillin, trimethoprim-sulfamethoxazole and amoxacilline-clavunic acid, while Gram-positive isolates showed high level of resistant to penicillin. On the other hand, meropenem, for Gram-negative and clindamycin for Gram-positive bacteria showed lower level of resistance. Multidrug resistance among Gram-positive and Gram-negative bacteria were (62.6%; 92/147) and (75.3%; 64/85), respectively with over all MDR (67.2%, 156/232).

## Conclusion

Inanimate objects commonly used by healthcare professionals are important sources of bacterial contamination. *S. aureus*, *S. epidermidis* and *E. coli* were the predominant isolates. Most Gram-negative bacteria were resistant to ampicillin while Gram-positive isolates showed high level of resistant to penicillin. All healthcare professionals should regularly disinfect their inanimate objects to prevent bacterial colonization and potential spread of infection.

## Introduction

Nosocomial infections are healthcare-associated infections (HAIs), that occur in patients during the process of acquiring healthcare services in a hospital or other healthcare facility, which was not present at the time of admission [1, 2]. Infection is frequently considered as HAI if it appears at or after 48 hours of admission and within 48–72 hrs after discharge [3]. Healthcare-associated infections may also be acquired by healthcare professionals during healthcare delivery [2]. The spread of HIAs is largely due to healthcare professionals' (HCPs) contaminated hands, their inanimate objects and contact with patients, inadequate equipment sterilization, and the rise of bacterial strains that are resistant to treatment [4]. Healthcare-associated infections to HCPs acquires during their day-to-day hospital activities such as specimen collection,

processing and discarding, handling and discarding of medical equipment, and contaminated objects as well as during direct contact with the patient at the time of examination [5].

Healthcare-associated infections are more concerning in the twenty-first century for a number of reasons. Among these are hospital housing a high number of sick patients, many of whom have compromised immune systems; an increase in outpatient care; numerous medical procedures that bypass the body's natural defenses; medical staff that moves between patients allowing pathogens to spread; poor hygiene guidelines for clothes, tools, cleaning, sterilizing, and other preventive actions; and the regular use of antimicrobial agents in hospitals, which exerts selection pressure towards the emergence of resistant strains of microorganisms [6].

Healthcare-associated infections are a major health concern for millions of people globally. Hospital settings and contaminated equipment are well-known sources of illness. Healthcare professionals may act as a mobile surface for transmission due to their contaminated inanimate objects [7]. Inanimate objects that are frequently touched by hands can act as reservoirs for infections that transfer to HCP hands and ultimately to patients. These inanimate objects used by healthcare professionals pick up harmful germs and disseminate the infection to others [7, 8]. Of these inanimate objects, mobile phones, gowns and stethoscopes are highly prone to bacterial contamination from the healthcare setting and are considered potential sources for these infections. *S. aureus*, *E. coli*, *Klebsiella pneumonia*, *Proteus* species, coagulase-negative staphylococcus species, *Acinetobacter* species and *P. aeruginosa*. These are the most frequent bacterial isolates from HCPs mobile phones, gowns and stethoscopes [9, 10].

Currently, mobile phones have become essential accessories for HCPs and social life [11, 12]. Healthcare professionals frequently utilize mobile phones in the hospital setting for internet browsing, infusion dose calculations, and electrolyte corrections in addition to communication [13, 14]. Despite all of the potential advantages, mobile phones are known to cause illnesses linked to HAI and play a significant role in becoming potential bacterial reservoirs [15, 16]. Mobile phones have the potential to spread nosocomial infections to other locations, like workers' home [17].

Healthcare professionals' gowns become contaminated with microorganisms through regular use, which may aggravate HAIs more likely [18]. Gowns are critical elements of personal protective equipment (PPE) since they are the second-most-used piece of PPE, following gloves [19, 20]. Gowns that serve as transmission vehicles have the potential to transmit pathogens from one patient to another [21, 22].

Similarly, the stethoscope can act as an important spreader of bacterial infection among HCPs and patients [23]. When the unclean parts of the stethoscope come into direct contact with the patient's skin and the physician's hand, commonly become colonized with pathogenic isolates and subsequently be transferred to other patients if the stethoscope is not disinfected [24, 25].

Healthcare-associated infections are more aggravated once a considerable number of microorganisms are resistant to conventional antimicrobials as well as to new drugs. There is a high risk of transmission of multidrug antibiotic-resistant microorganisms in hospital settings [26, 27]. In Ethiopia, HAIs among patients increased by more than twofold between 2009 and 2018 from 5.7% to 19.41%. This indicates that there is a significant concern for patients as well as HCPs, as HAIs may raise occupational risk among HCPs [28, 29]. Despite continuing efforts of hospital infection control, HAIs are still a major public health problem globally. There is no report about the role of gowns, mobile phones, and stethoscopes in the transmisions of healthcare associated infection in the study area. Therefore, this study aimed to assess bacterial profile, associated factors, and antimicrobial susceptibility patterns from inanimate objects used by healthcare professionals at Debre Markos Comprehensive Specialized Hospital, Northwest Ethiopia.

## Materials and methods

### Study area and setting

The study was conducted at Debre Markos Comprehensive Specialized Hospital, which is found in Debre Markos town, the capital of East Gojjam zone, located 302 km Northwest of Addis Ababa, the capital city of Ethiopia, and 264 km Southeast of Bahir Dar, the capital of Amhara National Regional State. It is one of the oldest public hospitals in the country which was established in 1957. It provides health services for approximately 255,248 patients per year from a catchment population of about 5 million people. The hospital consists of an outpatient department (OPDs), emergency ward, gynecology/obstetrics, maternity, medical, surgical, orthopedics, pediatric wards, operating room, intensive care unit (ICU), laboratory and pharmacy unites. It has more than 782 staffs, including both supportive and healthcare professionals. It gives service to East Gojjam, West Gojjam, Awi zone, and some parts of the Oromia region [30].

### Study design and period

A hospital-based cross-sectional study was conducted from April 10, 2023 to June 30, 2023.

### Source and study population

Inanimate objects used by healthcare professionals were the source populations. Similarily, their gowns, stethoscopes and mobile phones used by these healthcare professionals were the study populations during the study period.

### Dependent variable

Bacterial profiles of inanimate objects

### Independent variables

Socio-demographic characteristics includes such as age, sex, year of service, field of specialization, level of education, working wards, hygiene-related practice, hand washing, disinfecting of mobile phone, use of mobile phone at bed side for information, answering phone calls while attending patients, carry mobile phone with medical equipments, disinfecting of stethoscope, share stethoscope, frequency use of laundry for gown, infection prevention training and presence of infection prevention manual.

### Sample size determination and sampling techniques

The sample size was calculated by using single population proportion formula based on the assumption of a 95% confidence interval (Z $\alpha$/2 = 1.96), 5% margin of error, and prevalence of health care worker's fomites 57.6% from previous study conducted at Felege Hiwot Referral Hospital [9].

$$n = (Z\alpha/2)2\frac{P(1-P)}{d^2}$$

Where, n = the minimum sample size.

Z = the z-value at 95% confidence interval = 1.96;

P = the prevalence of bacteria from gowns, mobile phones and stethoscopes of HCP's = 57.6%.

d = margin of sampling error taken as 5% $(1.96)^2 x \frac{0.576(1-.576)}{(0.05)2} = 376$

Therefore, a total of 376 inanimate objects were included. This sample size was proportionally allocated to different healthcare professionals and working wards according to their

population size. Study participants were selected using simple random sampling technique after proportional allocation in each stratum of the occupational group. Gowns and mobile phones were collected from medical doctors, intern students, nurses, midwives, pharmacy professionals, medical laboratory professionals, radiologists, physiotherapists, ophthalmologists, dentists and dermatologists. Stethoscope were collected from health professionals who have stethoscopes by using a census until the required sample size was fulfilled because the allocated sample was larger than the physicians in the hospital (Fig 1).

### Eligibility criteria

Gowns, and mobile phones of all HCPs and stethoscopes of HCPs who have stethoscope at DMCSH were included in the study. Gowns, mobile phones, and stethoscopes from HCPs on maternal and annual leave during the study period were excluded.

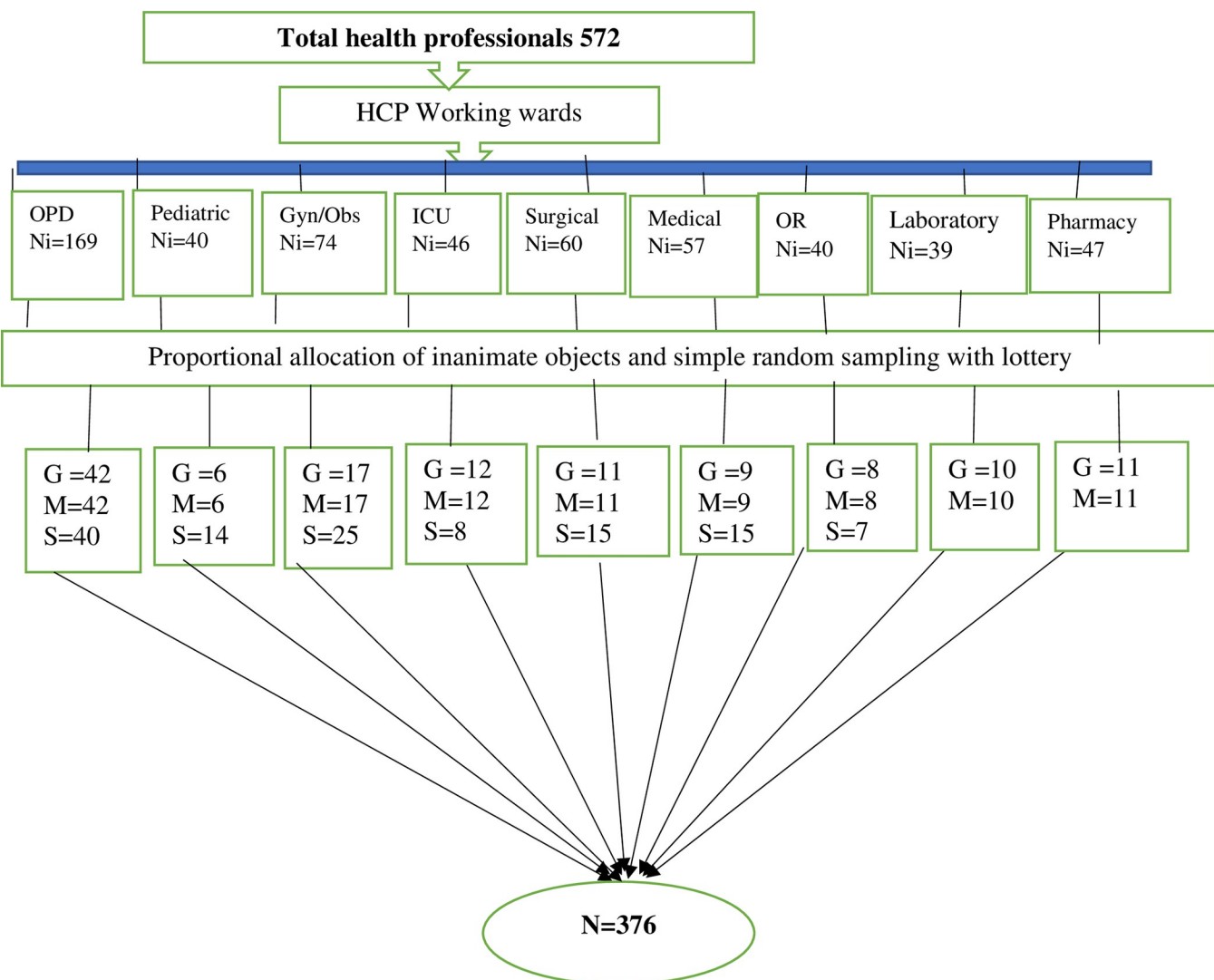

Key: - G=gown, M=mobile phone, S=stethoscope

**Fig 1. Sampling technique from healthcare professionals' inanimate objects used by healthcare professionals at DMCSH, Northwest Ethiopia, 2023.**

## Operational definitions

**Inanimate objects**: are gown, mobile phones and stethoscopes used by healthcare professionals at DMCSH.

**Hand washing**: refers to cleaning hands using alcohol- or hand sanitizers or washing hands with soap and water.

**Healthcare professionals**: includes medical doctors, anesthetists, nurses, midwives, medical intern students, radiologists, physiotherapists, psychiatrists, dermatologists, dentists, ophthalmologists, laboratory and pharmacy professionals who work at DMCSH.

**Multidrug-resistant (MDR)**: When a bacterial isolate is resistant to at least one drug in three or more classes of antimicrobial agents [31].

## Data collection

**Socio-demographic data collection.** Socio-demographic and other data related to healthcare professional's gowns, stethoscopes, and mobile phones were collected by self-administered structured questionnaires. A total of 376 inanimate objects wrere enroll from April 10, 2023 to June 30, 2023. Of these gowns, mobile phones and stethoscopes 126,126 and 124 swaps were collected, respectivly.

## Specimen collection and transportation

The data collector cleaned their hands using alcohol-based hand sanitizer before taking swab samples from HCP's inanimate objects. Powder-free disposable gloves were worn per sample during the sample collection process to prevent cross-contamination. The specimen was collected by using a sterile cotton swab moistened with saline from the gowns and mobile phones of all healthcare professionals, and stethoscope swabs were collected from healthcare professionals who have stethoscopes by using a census until the required sample size was fulfilled. Specifically, a swab from the cuff of the dominant hand and pocket mouth of the gown were collected. The entire surface of the diaphragm and earpieces of each stethoscope was swabbed. From mobile phones, swab was performed on fully stretched of the screen and backside of where the most frequent areas of contact with hands. Following collection, all swab specimens were inserted into a separate test tube and labeled with sample number, sample type, time, and date of sample collection. After collection, all swab samples were transported to DMCSH Microbiology laboratory and processed within 1hour of collection.

## Isolation and identification of bacteria

After the sample was aseptically collected, the moistened swab sample was inoculated onto blood agar, chocolate and MacConkey agar. All inoculated media was incubated at 35–37°C for 18 to 24 hrs. Additionally, inoculated chocolate agar media was incubated in 5% $CO_2$ enriched environment. Primary isolation of bacteria was made based on their colony characteristics, hemolytic pattern, Gram reaction and microscopic features. Then, further identification of bacterial organisms were analyzed with biochemical tests [32].

Biochemical tests were performed on colonies from pure cultures and inoculated onto different biochemical test media and incubated aerobically at 37°C for 18–24 hours for identification of the isolates. The following biochemical test was tested for Gram-negative bacteria: oxidase, indole, triple sugar iron agar, citrate, urea, lysine decarboxylase (LDC) agar, motility test, and mannitol test. For Gram-positive bacteria, a catalase test was done to differentiate staphylococcal from streptococcal isolates. A slide coagulase test was done to differentiate *S.*

*aureus* from coagulase-negative staphylococci (CoNS). Also manitol salt agar, and novobiocin test was done [20, 33].

## Antimicrobial susceptibility testing

The antimicrobial susceptibility testing were performed for all issolates by Kirby-Bauer disk diffusion technique on Muller-Hinton agar following the Clinical and Laboratory Standard Institute (CLSI) guideline. A loop full of bacteria (2–3 identical colonies) was taken by sterile wire loop from a pure culture colony and transferred to a test tube containing 5ml of normal saline, and the mixture was gently mixed to create a uniform suspension. The 0.5 McFarland standard was used to standardize the turbidity of the bacterial suspension [34].

Sterile cotton swabs were immersed into the suspension, remove excess fluid by gentle rotation of the swab to the inner surface of the tube, and inoculated the bacterial suspension over the entire surface of Mueller Hinton agar. The antimicrobial discs were placed and incubated at 37˚C for 18–24 hours. After the incubation period, the diameter zone of inhibition was measured and interpreted based on CLSI 2022 guidelines. The results were interpreted as resistant (R), intermediate (I) and sensitive (S) and bacterial isolates resistant to more than three classes if a drug was classified as multidrug-resistant (MDR) [34, 35]. These antibacterial discs were selected based on the pathogen, local availability, commonly prescribed antibiotics in the study area and CLSI, 2022 recommendations.

The following antimicrobials (Oxoid, LTD, UK) were tested for Gram-negative bacteria: ampicillin (AMP, 10μg), amoxicillin-clavulanic acid (AMC, 30μg), ceftriaxone (CRO, 30μg), ceftazidime (CAZ, 30μg), amikacin (AMK, 30μg), gentamicin (GEN, 10μg), meropenem (MEM, 10μg) ciprofloxacin (CIP, 5μg), trimethoprim/sulfamethoxazole (SXT, 1.25/23.75μg), tetracycline (TCY, 30μg), tobromycin (TOB, 10μg), cefuraxime (CXM, 30μg) and chloramphenicol (CHL, 30μg).

For Gram-positive bacteria: ciprofloxacin (CIP, 5μg), gentamicin (GEN, 10μg), oxacillin (OX, 5μg), azithromycin (AZM, 15μg), clindamycin (CLD, 2μg), trimethoprim/sulfamethoxazole (SXT, 1.25/23.75μg), penicillin (P, 10 units), cefoxitin (FOX, 30μg), chloramphenicol (CHL, 30μg) and doxycycline (DOX, 30μg) were tested [34].

## Data quality assurance

A structured questionnaire was prepared in a clear and precise way. The validation of the questionnaire was checked by doing a pre-test on 5% of the total sample size (19 clients) at Asrade Zewudie Memorial Hospital, Burie, Ethiopia. Data collectors were trained and standard operating procedures (SOPs), as well as manufacturer's instructions, were strictly followed during specimen collection, transportation and processing. Preparation of all culture media, and their sterility was checked by overnight incubation of 5% of the batch at 37˚C and observing for growth. Any growth in the culture medium was rejected, and replaced by new sterile batch. All prepared media and antibiotics were checked by inoculating standard reference strains, *S. aureus* (ATCC® 29213), *E. coli* (ATCC® 25922), *P. aeruginosa* (ATCC® 27853) and *S. pneumonia* (ACCT®49619) as quality control during the study period.

## Data processing and analysis

Data were entered by EPI-data -version 4.6 and exported to Statistical Package for Social Science (SPSS) program version 25 for analysis. Bivariable logistic regression was carried out to identify the associated factors with bacterial profile of gowns, stethoscopes, and mobile phones. Multivariate logistic regression analysis was fitted to variables that had a *P*-value of less than or equal to 0.25 in bivariable logistic regression. Finally, multivariate logistic regression analysis

was performed for adjusted odds ratio (AOR) with 95% confidence intervals. A *P*-value of ≤0.05 with 95%CI was considered statistically significant. The Hosmer and Lemeshow goodness of fit test was used for model fitness. Antimicrobial susceptibility tests were analyzed by using WHONET software.

### Ethical considerations

The data was collected after ethical clearance was obtained from the Institutional Research Ethics Review Committee (IRERC) of College of Medicine and Health Sciences, Debre Markos University (Reference number: HSC/RCS/127/11/12). A supporting letter was written from the College of Medicine and Health Sciences to DMCSH. Then, a written permission paper was obtained from DMCSH before data collection. All the study participants were informed about the aim of the research and written consent was obtained. Participants were free to continue or withdraw from the study. Confidentiality of the results were maintained.

## Results

### Socio-demographic characteristics of healthcare professionals

A total of 376 inanimate objects were swabbed from 191 healthcare professionals. Of these (63.9%; 122/191) were male HCPs. The mean age of the participants was 30.43 with SD ± 4.81 and (range 20 to 45) years. More than forty-two percent (42.9%; 82/191) of the HCPs were between the age of 30–34 years. Of the total HCPs who participated in this study, most of them (59.2%; 113/191) had less than 5 years of service. With respect to field of specialization, most of the participants (25.7%; 49/191) were nurse professionals. Regarding working wards (31.9%; 61/192), (16.8%; 32/191), and (9.9%; 19/191) HCPs' inanimate objects were from OPD, GYN/Obs and surgical ward, respectively. From the total inanimate objects, (33.5%; 126/376), (33.5%; 126/376), and (33%; 124/376) were mobile phones, gowns and stethoscopes, respectively (**Table 1**).

### Hygiene-related practice of healthcare professionals

The overall HCPs' practice towards regular hand washing and disinfection of their inanimate objects were presented in Table 2. The majority of the HCPs did not regularly wash their hands after touching a patient (79.6%; 152/191). Moreover, (61.1%; 77/126) of the HCPs did not regularly disinfect their mobile phones. Many participants were (72.2%; 91/126) used their mobile phones at bedside for communication and medical information.

### Prevalence of bacterial contamination

Overall, 60.1% (226/376) (95%CI: 55.1–65) of the inanimate objects were contaminated with different bacteria from 376 objects among 191 healthcare professionals. Mobile phones had the highest percentage of bacterial contamination (65.1%; 82/126), followed by gowns (60.3%; 76/126) and stethoscopes (54.8%; 68/124). Study participants whose inanimate object was mobile phone (70.2%; 40/57) and gown (66.1%; 39/59) had less than 5 years of service, whereas, participants with stethoscope (63.0%; 17/27) had 5–10 years of service and were a high rate of bacterial contamination. By profession, the contamination rate of inanimate object was (90%; 9/10) laboratory professionals' mobile phone, (80%; 12/13) intern students' gown and (73.3%; 11/20) nurses' stethoscopes. The highest frequency of inanimate object contamination with bacteria was observed among HCPs working in intensive care unit and neonatal care unit (ICU &NICU) (87.5%; 7/8/) stethoscope, and (83.3; 10/12) gown and (90.0%; 9/10) mobile phone from laboratory unit (**Table 3**).

**Table 1. Socio-demographic characteristic of HCPs from inanimate objects used by healthcare professionals at DMCSH, Northwest Ethiopia 2023.**

| Variables | | Frequency (N) | Percentage (%) |
|---|---|---|---|
| Sex | Male | 122 | 63.9 |
| | Female | 69 | 36.1 |
| Age (in years) | 20–24 | 17 | 8.9 |
| | 25–29 | 56 | 29.3 |
| | 30–34 | 82 | 42.9 |
| | >35 | 36 | 18.8 |
| Year of service | <5 year | 113 | 59.2 |
| | 5–10 year | 55 | 28.8 |
| | >10 year | 23 | 12.0 |
| Field of specialization | Nurse | 49 | 25.7 |
| | Lab. Professional | 10 | 5.2 |
| | Pharm. Professional | 11 | 5.8 |
| | Medical Doctor | 40 | 20.9 |
| | Midwifery | 11 | 5.8 |
| | Intern students | 61 | 31.9 |
| | Others* | 9 | 4.7 |
| Level of education | Diploma | 11 | 5.8 |
| | BSc | 74 | 38.7 |
| | MD | 26 | 13.6 |
| | Med. intern students | 61 | 31.9 |
| | Specialist | 19 | 9.9 |
| Wards | OPDs | 61 | 31.9 |
| | Pediatrics | 15 | 7.9 |
| | Gynecology/ Obs. | 32 | 16.8 |
| | ICU & NICU | 15 | 7.9 |
| | Laboratory unit | 10 | 5.2 |
| | OR | 10 | 5.2 |
| | Surgical | 19 | 9.9 |
| | Medical | 18 | 9.4 |
| | Pharmacy unit | 11 | 5.8 |
| Inanimate object | Gown | 126 | 33.5 |
| | Mobile phone | 126 | 33.5 |
| | Stethoscope | 124 | 33.0 |

*Physiotherapy, X-ray technicians, anesthesia, ophthalmology profession, psychiatry and dentistry; MD: Medical Doctor; ICU&NICU: Intensive care unit and Neonatal intensive care unit; OPD: Outpatient department; OR: Operation room

## Bacterial isolates from HCPs inanimate objects

Out of 376 swab samples processed, (61.7%; 232/376) bacteria were isolated from 191 study participants. Only six inanimate objects showed co-infection with two bacterial species. From the total bacterial isolates (63.4%; 147/232) and (36.6%; 85/232) were Gram-positive and Gram-negative bacteria, respectively. Overall, *S. aureus* was the predominant isolate accounting (22.1%; 83/376) followed by *S. epidermidis* (17.0%, 64/376), *E. coli* (8.8%, 33/376) and *P. aeruginosa* (4.9%; 18/376). *S. aureus* was the predominant isolate from gown (39.2%; 31/83) and from mobile phone (41%; 34/83). On the other hand, *S. epidermidis* was the most frequent isolate from stethoscope (35.9%; 23/64) (**Fig 2**).

**Table 2.** Hygiene-related practice of HCPs from inanimate objects used by healthcare professionals at DMCSH, Northwest Ethiopia 2023.

| Variables | Frequency | |
|---|---|---|
| | Yes N(%) | No N(%) |
| Regular hand washing before attending the patient | 135(70.7) | 56(29.3) |
| Regular hand washing after touching a patient | 39(20.4) | 152(79.6) |
| Regular use antiseptic for hands | 113(59.2) | 78(40.8) |
| Regular hand washing after body fluid exposure | 179(93.7) | 12(6.3) |
| Regular hand washing before and after clean/aseptic procedure | 153(80.1) | 38(19.9) |
| Regular disinfection of mobile phone | 49(38.9) | 77(61.1) |
| Use of mobile phone at bed side for medical information | 90(71.4) | 36(28.6) |
| Wash hands after using a mobile phone in the hospital | 35(27.8) | 91(72.2) |
| Carry mobile phone with patient care material | 82(65.1) | 44(34.9) |
| Answer phone calls while attending patients | 62(49.2) | 64(50.8) |
| Regularly disinfection of stethoscope after examining each patient | 43(34.7) | 81(65.3) |
| Share stethoscope | 34(27.4) | 90(72.6) |
| Use laundry for your gown/ clean your gown regularly | 91(72.2)) | 35(27.8) |
| Think that gown, mobile phones and stethoscope can carry bacteria | 182(95.3) | 9(4.7) |
| Taking any training in infection prevention | 88(46.1) | 103(53.9) |
| Infection prevention manual in working area | 153(80.1) | 38(19.9) |

## Factors associated with contamination of healthcare professionals' inanimate objects

From bi-variable analysis age, level of education, working ward, regularly washing hands before attending patient, hand washing after touching patient, regular use of antiseptic for hands, regular hand washing before and after clean/aseptic procedure, regular disinfection of mobile phone, use of mobile phone at bedside to medical information, carry mobile phone with medical equipments and during patient care, regular disinfection of stethoscope after each patient, share stethoscope, use laundry for cleaning gown regularly, infection prevention training and presence of infection prevention manual were candidates for multivariate analysis ($P \leq 0.25$).

In multivariate analysis, inanimate objects of HCPs from Gyn/Obs ward (AOR: 8.69; 95% CI: 1.09–69.41, $P = 0.041$), no regular disinfection of mobile phones (AOR: 2.69; 95%CI: 1.15–6.25; $P = 0.021$), carry mobile phones medical equipments and during patient care (AOR: 2.72; 95%CI: 1.18–6.29, $P = 0.019$), no regular disinfection of stethoscopes after examining each patient (AOR: 3.06, 95% CI: 1.23–7.59; $P = 0.016$) and no taking infection prevention training (AOR: 3.91; 95%CI:1.71–8.93; $P = 0.001$) were associated with bacterial contamination of HCPs inanimate objects (**Tables 4 and 5**).

## Antimicrobial susceptibility pattern

Among Gram-negative bacterial isolates, *E. coli* (n = 33) was highly resistance to ampicillin and (97%, 32/33) and amoxacilline-clavunic acid and trimethoprim-sulfamethoxazole each accounted (93.9%, 31/33). While, meropenem (100%) and ciprofloxacin (90.9%) were sensitive against *E. coli* isolates. On the other hand, *Klebsiella* sppecies showed high resistance rate to ampicillin, trimethoprim-sulfamethoxazole and amoxacilline-clavunic acid each accounts (92.3%; 12/13). However, *Klebsiella* spcies showed lower resistance rate against meropenem, ciprofloxacin and gentamicin, (0%, 0/13), (15.4%, 2/13) and (30.8%, 4/13), respectively.

**Table 3. Bacterial contamination of inanimate objects used by healthcare professionals at DMCSH, Northwest Ethiopia 2023.**

| Characteristics | | Contamination status | | | | | |
|---|---|---|---|---|---|---|---|
| | | Gown | | Mobile phone | | Stethoscope | |
| | | Contaminated N (%) | Non-contaminated N(%) | Contaminated N (%) | Non-contaminated N(%) | Contaminated N (%) | Non-contaminated N(%) |
| Sex | Male | 49(60.5) | 32(39.5) | 52(64.2) | 29(35.8) | 46(55.4) | 37(44.6) |
| | Female | 27(60.0) | 18(40.0) | 30(66.7) | 15(33.3) | 22(53.7) | 19(46.3) |
| Age (in years) | 20–24 | 7(41.2) | 10(58.8) | 10(55.6) | 8(44.4) | 4(66.7) | 2(33.3) |
| | 25–29 | 22(64.7) | 12(35.3) | 19(54.3) | 16(45.7) | 22(56.4) | 17(43.6) |
| | 30–34 | 33(67.3) | 16(32.7) | 35(71.4) | 14(28.6) | 25(45.5) | 30(54.5) |
| | >35 | 14(53.8( | 12(46.2) | 18(75.0) | 6(25.0) | 17(70.8) | 7(29.2) |
| Year of service | <5 year | 39(66.1) | 20(33.9) | 40(70.2) | 17(29.8) | 47(52.2) | 43(47.8) |
| | 5–10 year | 27(57.4) | 20(42.6) | 29(60.4) | 19(39.6) | 17(63) | 10(37) |
| | >10 year | 10(50.0) | 10(50.0) | 13(61.9) | 8(38.1) | 4(57.1) | 3(42.9) |
| level of education | Diploma | 6(54.5) | 5(45.5) | 6(54.5) | 5(45.5) | 0 | 0 |
| | BSc | 39(56.5) | 30(43.5) | 42(60.9) | 27(39.1) | 13(59.1) | 9(40.9) |
| | MD | 10(62.5) | 6(37.5) | 11(68.8) | 5(31.3) | 12(48.0) | 13(52.0) |
| | Med. intern students | 12(80.0) | 3(20.0) | 11(73.3) | 4(26.7) | 30(49.2) | 31(50.8) |
| | Specialist | 9(60.0) | 6(40.0) | 12(80.0) | 3(20.0) | 13(81.3) | 3(18.8) |
| Field of specialization | Nurses | 26(57.8) | 19(42.2) | 26(57.8) | 19(42.2) | 11(73.3) | 9(26.7) |
| | Lab. Professionals | 6(60.0) | 4(40.0) | 9(90.0) | 1(10.0) | 0 | 0 |
| | Pharm. professionals | 4(36.4) | 7(63.6) | 5(45.5) | 6(54.5) | 0 | 0 |
| | Doctor | 17(65.4) | 9(34.6) | 20(76.9) | 6(23.1) | 23(59.0) | 16(41.0) |
| | Midwifery | 7(70.0) | 3(30.0) | 5(50.0) | 5(50.0) | 2(50.0) | 2(50.0) |
| | Intern students | 12(80.0) | 3(20.0) | 11(73.3) | 4(26.7) | 30(49.2) | 31(50.8) |
| | Others | 4(44.4) | 5(55.6) | 6(66.7) | 3(33.3) | 2(40) | 3(60) |
| Wards | OPDs | 21(50) | 21(50) | 25(59.5) | 17(40.5) | 22{55) | 18(45) |
| | Pediatrics | 3(50) | 3(50) | 1(16.7) | 5(83.3) | 6(42.9) | 8(57.1) |
| | Gynecology/ Obs. | 14(82.4) | 3(17.6) | 12(70.6) | 5(29.4) | 13(52) | 12(48) |
| | ICU & NICU | 10(83.3) | 2(16.7) | 9(75) | 3(25) | 7(87.5) | 1(12.5) |
| | Laboratory unit | 6(60) | 4(40) | 9(90) | 1(10) | 0 | 0 |
| | OR | 5(62.5( | 3(37.5) | 6(75) | 2(25) | 4(57.1) | 3(42.9) |
| | Surgical | 6(54.5) | 5(45.5) | 7(63.6) | 4(36.4) | 8(53.3) | 7(46.7) |
| | Medical | 7(77.8) | 2(22.2) | 8(88.9) | 1(11.1) | 8(53.3) | 7(53.3) |
| | Pharmacy unit | 4(36.4) | 7(63.6) | 5(45.5) | 6(54.5) | 0 | 0 |

OPD: Outpatient department, ICU & NICU: Intensive care unit&neonatal intensive care unit, OR: Operation room, MD: Medical Doctor, BSc: Bachelor of Science

Similarly, *P. aeruginosa* showed a higher resistance rate against piperacillin (94.4%,17/18), cefetazidime (72.2%, 13/18) and ciprofloxacin (66.6%, 12/18) and lower resistance rate to meropenem (11.1%; 2/18) (**Table 6**).

*S. aureus* showed high sensitivity to clindamycin (91.6%, 76/83) and gentamycin (84.3%, 70/83). However, it was highly resistant to penicillin (97.6%, 81/83), trimethoprim-sulfamethoxazole (67.5%, 56/83), azithromycin (50.6%, 42/83) and cefoxitin (36.1%, 30/83). On the other hand, *S. epidermidis* was resistant to penicillin (98.4%, 63/64), trimethoprim-sulfamethoxazole (65.6%, 42/64), and doxycycline (48.4%, 31/64). Based on cefoxitin resistance (36.1%, 30/83) was methicillin-resistant *S. aureus* (MRSA) and (63.9%), 53/83) and methicillin-sucsceptible *S. aureus* (MSSA) (**Table 7**).

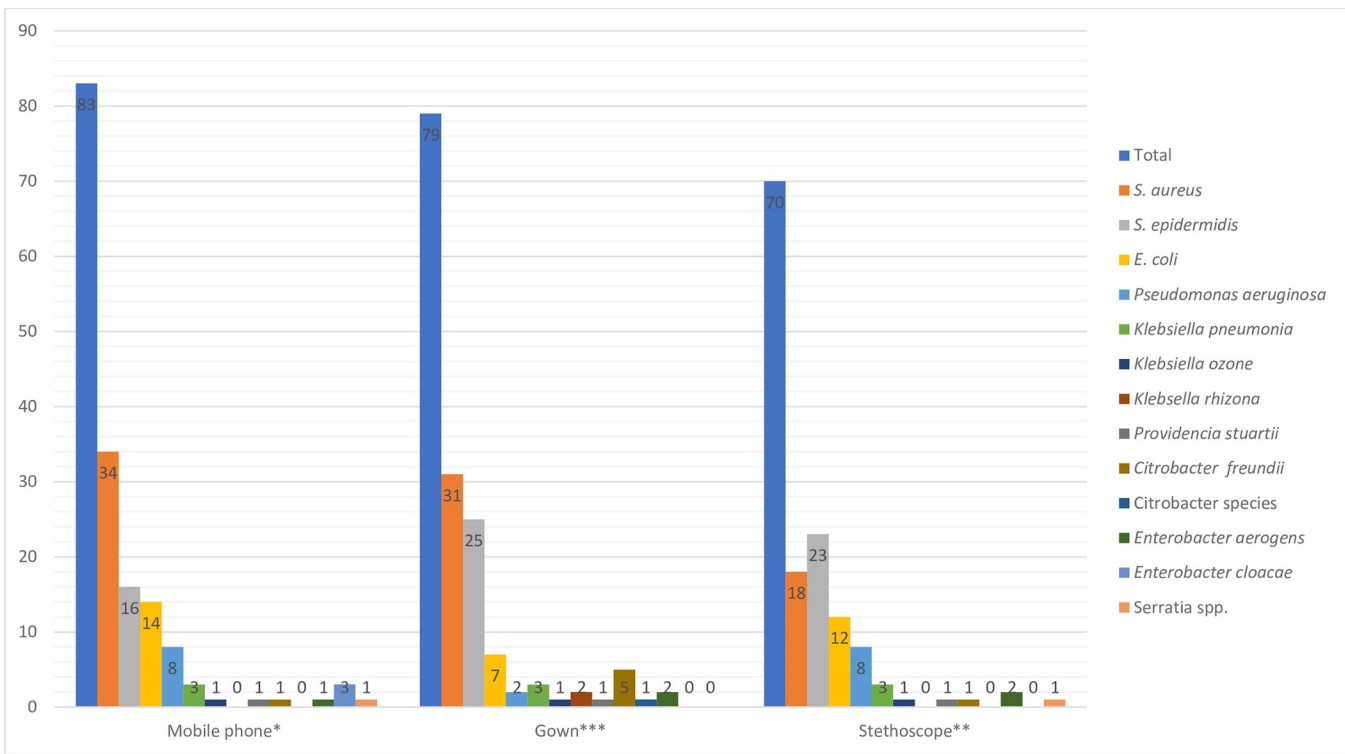

***: Three gowns co-infected with two bacteria, **: One stethoscope co-infected with two bacteria, *: One mobile phone co-infected with two bacteria

**Fig 2. Proportion of bacterial isolates from healthcare professionals' inanimate objects used by healthcare professionals at DMCSH, Northwest Ethiopia, 2023.**

### Multidrug resistance profiles of the isolates

Multidrug resistance among Gram-positive and Gram-negative bacteria isolates were (62.6%; 92/147) and (75.3%; 64/85), respectively with over all MDR (67.2%, 156/232). Among all bacteria isolates *E. coli* (72.7%; 24/33) predominantly showed MDR followed by *Klebsiella* species (76.9%; 10/13), *P. aeruginosa* (66.7%;12/18), *S. epidermidis* (67.2%; 43/64), and *S. aureus* (63.9%; 53/83). From Gram-negative bacteria, *E. coli* showed MDR among 28 isolates (72.7%; 24/33) isolates. Similarly, *Klebsiella* species showed MDR among (76.9%; 10/13) (**Table 8**). *P. aeruginosa* were resistance to three or more antimicrobials among 12 isolates (66.7%; 12/18) (**S1 Table**). Of Gram-positive isolates, *S. epidermidis* was multidrug resistance among 46 isolates (71.8%; 46/64). Similarly, *S. aureus* was MDR among 53 isolates (63.8%; 53/83) isolates (**S2 Table**).

### Discussions

Inanimate objects including stethoscopes, mobile phones, and gowns are essential materials in HCPs' day-to-day activities throughout the world. These inanimate objects are used by healthcare professionals' are thought to be possible sources of diseases related to healthcare because they collect dangerous germs and spread the infection to others. They are also particularly susceptible to bacterial contamination from the hospital context [36].

The overall prevalence of bacterial contamination of HCPs' inanimate objects was (60.1%; 95%CI: 55.1–65%) which was in line with result reported in Bahir Dar, Ethiopia (57.6%) [9]. However, the current study was lower than a study conducted in Harar, Ethiopia (94.2%) [37],

**Table 4. Socio-demographic factors associated with contamination of healthcare professionalss inanimate objects used by HCPs at Debre Markos Comprehensive Specialized Hospital, Northwest Ethiopia 2023.**

| Variables | | Inanimate object contamination | | COR (95% CI) | P-value | AOR(95%CI) | P-value |
|---|---|---|---|---|---|---|---|
| | | Yes (%) | No (%) | | | | |
| Age | 20–24 | 10(55.6) | 8(44.4) | 0.42(0.11–1.55) | 0.190 | 0.61(0.09–3.71) | 0.591 |
| | 25–29 | 19(54.3) | 16(45.7) | 0.39(0.13–1.24) | 0.111 | 0.36(0.07–2.02) | 0.25 |
| | 30–34 | 35(71.4) | 14(28.6) | 0.83(0.27–2.54) | 0.748 | 1.31(0.266–6.46) | 0.74 |
| | >35 | 18(75.0) | 6(25.0) | 1 | | | |
| Year of service | < 5 years | 39(66.1) | 20(33.9) | 1.95(0.69–5.46) | 0.203 | 0.89(0.24–3.42) | 0.876 |
| | 5–9 years | 27(57.4) | 20(42.6) | 1.35(0.47–3.86) | 0.575 | 0.77(0.196–2.99) | 0.702 |
| | >10 years | 10(50.0) | 10(50.0) | 1 | | | |
| Level of education | Diploma | 6(54.5) | 5(45.5) | 0.80(0.17–3.86) | 0.781 | 1.45(0.17–9.72) | 0.701 |
| | BSc | 39(56.5)) | 30(43.5) | 0.87(0.28–2.70) | 0.805 | 0.67(0.18–2.52) | 0.550 |
| | MD | 10(62.5) | 6(37.5) | 1.11(0.26–4.72) | 0.886 | 1.74(0.32–9.45) | 0.524 |
| | Med. Intern stud. | 12(80.0) | 3(20.0) | 2.67(0.52–13.66) | 0.239 | 1.82(0.24–13.64) | 0.558 |
| | Specialist | 9(60.0) | 6(40.0) | 1 | | | |
| Working ward | OPD | 21(50.0) | 21(50.0_ | 1.75(0.45–6.88) | 0.423 | 1.85(0.37–9.27) | 0.454 |
| | Pediatrics | 3(50.0) | 3(50.0) | 1.75(0.23–13.16) | 0.587 | 1.65(0.16–17.22) | 0.675 |
| | GYN/OBS | 14(82.4) | 3(17.6) | 8.17(1.42–47.02) | 0.019 | 8.69(1.09–69.41) | 0.041* |
| | ICU & NICU | 10(83.3) | 2(16.7) | 8.75(1.24–61.68) | 0.029 | 8.82(0.95–81.45) | 0.055 |
| | Laboratory unit | 6(60.0) | 4(40.0) | 2.63(0.45–15.31) | 0.283 | 1.84(0.26–12.91) | 0.539 |
| | OR | 5(62.5) | 3(37.5) | 2.92(0.44–19.23) | 0.266 | 3.19(0.34–29.74) | 0.309 |
| | Surgical ward | 6(54.5) | 5(45.5) | 2.10(0.38–11.59) | 0.395 | 2.29(0.29–17.75) | 0.427 |
| | Medical ward | 7(77.8) | 2(22.7) | 6.13(0.83–45.02) | 0.075 | 4.37(0.42–45.18) | 0.216 |
| | Pharm. Unit | 4(36.4) | 7(63.6) | 1 | | | |

Key: BSc: Bachelor of Science, MD: Medical doctor, OR: Operation theatre, OPD: Outpatient department, GYN/OBS: Gynecology/obstetrics, ICU&NICU: Intensive care unit and Neonatal Intensive care unit

Nigeria (77.7%) [38], Tanzania (73.3%) [21] and another study from Nigeria (65.5%) [39]. On the other hand, this is higher than result reported in Pakistan (43%) [40], Saudi Arabia (30%) [41] and Rajshahi 19% [42]. The observed variation may have resulted from variations in the adherence to hygiene-related practices, regular hand washing practices, HCPs' awareness of the role of inanimate objects play in microbial transmission, variations in the regular laundry use gown habits, and storage.

In the present study, the prevalence of bacterial contamination of HCPs' mobile phones was 65.1% (56.1–73.4%, 95%CI), which was similar to a study conducted in Addis Ababa, Ethiopia (62%) [43], Jimma (71.2%) [44]. However, it was lower compared to a research conducted in Gondar, Ethiopia (98%) [45], India (100%) [46], Italia (100%) [47], China (95.5%) [48] and in Sana'a city, Yemen (70%) [49]. On the other hand, the current study was higher compared to previous report from Saudi Arabia (36.9%) [50] and Egypt (46.3%) [51]. The differences in the study HCPs adherence to infection prevention, mobile phone usage patterns, variation in regular disinfecting of mobile phones, and personal behaviors, such as picking noses and touching their phones, could be contributing factors to the variation in the contamination.

In this study, the prevalence of bacterial contamination of HCPs' gown was 60.3% (51.6–68.5%, 95%CI). It was in line with a study conducted in Bahir Dar (55.7%) [9]. However, it was lower than studies conducted in Tanzania (73.3%) [21] and Nigeria (77.7%) [38]. But higher

**Table 5. Hygiene-related factors associated with contamination of HCPs inanimate objects used by HCPs at Debre Markos Comprehensive Specialized Hospital, Northwest Ethiopia 2023.**

| Variables | | Inanimate object contamination | | COR (95% CI) | P-value | AOR(95% CI) | P-value |
|---|---|---|---|---|---|---|---|
| | | Yes (%) | No (%) | | | | |
| Hand washing before attending patient | Yes | 54 (61.4) | 34 (38.6) | 1 | | | |
| | No | 14 (38.9) | 22 (61.1) | 0.40(0.18–0.89) | 0.024 | 0.43(0.17–1.11) | 0.081 |
| Hand washing after touching patient | Yes | 20 (57.1) | 15 (42.9) | 1 | | | |
| | No | 62 (68.1) | 29 (31.9) | 1.60(0.72–3.57 | 0.248 | 1.56(0.47–5.19) | 0.469 |
| Regularly use antiseptic for hands | Yes | 43 (62.3) | 26 (37.7) | 1 | | | |
| | No | 25 (45.5) | 30 (54.5) | 0.50(.25–1.04) | 0.062 | 0.43(.18–1.05) | 0.065 |
| Regular hand washing before and after clean/aseptic procedure | Yes | 56 (60.9) | 36 (39.1) | 1 | | | |
| | No | 26 (76.5) | 8(23.5) | 2.09(.85–5.12) | 0.107 | 2.14(0.63–7.31) | .0.223 |
| Regular disinfection of mobile phone | Yes | 23 (46.9) | 26 (53.1) | 1 | | | |
| | No | 59 (76.6) | 18 (23.4) | 3.71(1.715–8.01) | 0.001 | 2.69(1.15–6.25) | 0.021* |
| Use mobile phone for medical information | Yes | 54 (60.0) | 36 (40.0) | 0.43(0.18–1.05) | 0.063 | 0.63(0.15–2.69) | 0.534 |
| | No | 28 (77.8) | 8(22.2) | 1 | | | |
| Carry mobile phone with patient care equipments | Yes | 48 (77.4) | 14 (22.6) | 3.03 (1.39–6.55) | 0.005 | 2.72(1.18–6.29) | 0.019* |
| | No | 34 (53.1) | 30 (46.9) | 1 | | | |
| Use laundry for your gown/ clean your gown regularly | Yes | 51 (56.0) | 40 (44.0) | 1 | | | |
| | No | 25 (71.4) | 10 (28.6) | 1.96(0.85–4.55) | 0.117 | 2.45(0.99–6.07) | 0.052 |
| Regular disinfection of stethoscope after examining each patient | Yes | 16 (37.2) | 27 (62.8) | 1 | | | |
| | No | 52 (64.2) | 29 (35.8) | 3.03(1.41–6.52) | 0.005 | 3.06(1.23–7.59) | 0.016* |
| Share stethoscope | Yes | 12 (35.3) | 22 (64.7) | 0.33 (0.15–0.75) | 0.008 | 0.56(0.18–1.75) | 0.317 |
| | No | 56 (62.2) | 34 (37.8) | 1 | | | |
| Infection prevention training | Yes | 20 (35.1) | 37 (64.9) | 1 | | | |
| | No | 48 (71.6) | 19 (28.4) | 4.67(2.19–9.99) | 0.000 | 3.91(1.71–8.93) | 0.001* |
| Infection prevention manual | Yes | 50 (56.2) | 39 (43.8) | 1 | | | |
| | No | 26 (70.3) | 11 (29.7) | 1.84(0.81–4.19) | 0.144 | 1.46(0.56–3.81) | 0.443 |

**Table 6. Antimicrobial susceptibility patterns of Gram-negative isolates from inanimate objects used by healthcare professionals at DMCSH, Northwest Ethiopia 2023.**

| Gram-negative bacteria | ASP | Antimicrobials with susceptibility profile N (%) | | | | | | | | | | |
|---|---|---|---|---|---|---|---|---|---|---|---|---|
| | | AMK | CAZ | PIP | MEM | GEN | CIP | TOB | | | |
| *P. aeruginosa* (n = 18) | S | 14(77.8) | 5(27.8) | 0 | 16(88.9) | 8(44.4) | 6(33.3) | 9(50) | | | |
| | I | 0 | 0 | 1(5.6) | 0 | 0 | | 0 | | | |
| | R | 4(22.2) | 13(72.2) | 17(94.4) | 2(11.1) | 10(55.6) | 12(66.7) | 9(50) | | | |
| | | AMP | AMC | CXM | CAZ | CRO | MEM | GEN | CIP | SXT | CHL | TCY |
| *Klebsiella* spp. (n = 13) | S | 0 | 1(7.7) | 5(38.5) | 7(53.8) | 6(46.2) | 13(100) | 9(69.2) | 11(84.6) | 0 | 7(53.8) | 5(38.5) |
| | I | 1(7.7 | 0 | 1(7.7) | 0 | 1(7.7) | 0 | 0 | 0 | 1(7.7) | 0 | 0 |
| | R | 12(92.3) | 12(92.3%) | 7(53.8) | 6(46.2) | 6(46.2) | 0 | 4(30.8) | 2(15.4) | 12(92.3) | 6(46.2)) | 8(61.5) |
| *Enterobacter* spp. (n = 8) | S | 1(12.5) | 1(12.5) | 4(50) | 5(62.5) | 4(50) | 8(100) | 7(87.5) | 7(87.5) | 0 | 5(62.5) | 4(50) |
| | I | 0 | 1(12.5) | 0 | 0 | 0 | 0 | 0 | 0 | 1(12.5) | 0 | 0 |
| | R | 7(87.5) | 6(75) | 4(50) | 3(37.5) | 4(50) | 0 | 1(12.5) | 1(12.5) | 7(87.5) | 3(37.5) | 4(50) |
| *Citrobacter* spp. (n = 8) | S | 0 | 0 | 2(25) | 5(62.5) | 5(62.5) | 8(100) | 5(62.5) | 7(87.5) | 0 | 3(37.5) | 1(12.5) |
| | I | 1(12.5) | 1(12.5) | | 1(12.5) | 0 | 0 | 0 | 0 | 1(12.5) | 0 | 1(12.5) |
| | R | 7(87.5) | 7(87.5) | 6(75) | 2(25) | 3(37.3) | 0 | 3(37.3) | 1(12.5) | 7(87.5) | 5(62.5) | 6(75) |
| *Providencia* spp. (n = 3) | S | 0 | 1(33.3) | 1(33.3) | 2(66.7) | 2(66.7) | 3(100) | 3(100) | 3(100) | 0 | 1(33.3) | 1(33.3) |
| | I | 1(33.3) | 0 | 1(33.3) | 0 | 1(33.3) | 0 | 0 | 0 | 1(33.3) | 0 | 0 |
| | R | 2(66.7) | 2(66.6) | 1(33.3)) | 1(33.3) | 0 | 0 | 0 | 0 | 2(66.7) | 2(66.7) | 2(66.7) |
| *Serratia* (n = 2) | S | 0 | 0 | 1(50) | 1(50) | 2(100) | 2(100) | 1(50) | 2(100) | 1(50) | 01(50) | 1(50) |
| | I | 0 | 1(50) | 0 | 1(50) | 0 | 0 | 0 | 0 | 0 | 0 | 0 |
| | R | 2(100) | 1(50) | 1(50) | 0 | 0 | 0 | 1(50) | 0 | 1(50) | 1(50) | 1(50) |
| *E.coli* (n = 33) | S | 0 | 1(3) | 13(39.4) | 24(72.2) | 19(57.6) | 33(100) | 27(81.8) | 30(90.9) | 0 | 16(48.5) | 10(30.3) |
| | I | 1(3) | 1(3) | 1(3) | 1(3) | 0 | 0 | 0 | 1(3) | 2(6.1) | 0 | 1(3) |
| | R | 32(97) | 31(93.9) | 19(57.6) | 8(24.2) | 14(42.4) | 0 | 6(18.2) | 2(6.1) | 31(93.4) | 17(51.5) | 22(66.7) |

Key: ASP: Antimicrobial susceptibility pattern, R: resistant, I: Intermediate, S: susceptible, AMK: Amikacin, AMP: Ampicillin, CRO: Ceftriaxone, CHL: Chloramphenicol, CIP: Ciprofloxacin, TCY: Tetracycline, GEN: Gentamicin, SXT: Sulfamethoxazole/trimetoprim, AMC: Amoxacilline-clavunic acid, CXM: Cefuroxime, CAZ: Ceftazidime, MEM: Meropenem, PIP: Pipracin, TOB: Tobromycin

than studies conducted in Nepal (50%) [52] and America (0–16%) [53]. This discrepancy might be due to differences in the use of regular laundry gowns (practice of cleaning gowns).

The prevalence of bacterial contamination of HCPs stethoscopes was 54.8% (45.7–63.8%, 95%CI). This was in line with study conducted in Addis Ababa, Ethiopia (53.8%) [54] and

**Table 7. Antimicrobial susceptibility patterns of Gram-positive isolates from inanimate objects used by healthcare professionals at DMCSH, Northwest Ethiopia 2023.**

| Gram-positive bacteria | ASP | Antimicrobials susceptibility profile N (%) | | | | | | | | | |
|---|---|---|---|---|---|---|---|---|---|---|---|
| | | PEN | FOX | OX | GEN | CIP | SXT | CLN | AZM | CHL | DOX |
| *S. aureus* (n = 83) | S | 2(7.2) | 53(63.9) | ND | 70(84.3) | 60(72.3) | 18(21.7) | 76(91.6) | 35(42.2) | 58(69.9) | 44(53) |
| | I | - | - | ND | 1(1.2) | 3(3.6) | 6(7.2) | 1(1.2) | 6(7.2) | 2(2.4) | 8(9.6) |
| | R | 81(97.6) | 30(36.1) | ND | 12(14.5) | 20(24.1) | 56(67.5) | 6(7.2) | 42(50.6) | 23(27.7) | 31(37.3) |
| *S. epidermidis* (n = 64) | S | 1(1.6) | ND | 40(62.5) | 52(81.3) | 36(56.3) | 20(31.2) | 56(87.5) | 29(45.3) | 48(75) | 27(42.2) |
| | I | - | ND | - | 0 | 7(10.9) | 2(3.1) | 0 | 4(6.3) | 0 | 6(9.3) |
| | R | 63(98,4) | ND | 24(37.5) | 12(18.9) | 21(32.8) | 42(65.6) | 8(12.5) | 31(48.4) | 16(25) | 31(48.4) |

Key: ASP: Antimicrobial susceptibility pattern, CHL: Chloramphenicol, PEN: Penicillin, CIP: Ciprofloxacin, CN: Gentamicin, FOX: Cefoxitin, SXT: Sulfamethoxazole/trimetoprim, CLN: Clindamycin, DOX: Doxycycline AZM: Azithromycin, OX: Oxacillin, S: susceptible, R: resistant, I: intermediate, ND: Not done

**Table 8. Multidrug resistance pattern of Gram-negative isolates from inanimate objects used by healthcare professionals at DMCSH, Northwest Ethiopia 2023.**

| Name of Antimicrobial | No of classes non susceptible | *E. coli* (n = 33) n (%) | *Klebsiella* spp. (n = 13) n (%) | *Enterobacter* spp. (n = 8) n (%) | *Citrobacter* spp. (n = 8) n (%) | *Providencia* spp. (N = 3) n (%) | *Serratia* spp. (n = 2) n (%) |
|---|---|---|---|---|---|---|---|
| None | | | | | | 1(33.3) | |
| AMP | | | | | | | 1(50) |
| AMP*, AMC* | 1 | 2(6.1) | 1(7.7) | 1(12.5) | | | |
| AMP, SXT | | 2(6.1) | 1(7.7) | 1(12.5) | | - | |
| AMC, SXT | | 1(3) | 1(7.7) | 1(12.5) | 1(12.5) | - | - |
| AMP,TCY | | - | | | 1(12.5) | | |
| AMP, AMC, SXT | 2 | 4(12.1) | - | | | - | - |
| AMP, AMC, CXM, CIP, SXT | 4 | 2(6.1) | 1(7.7) | | | | |
| AMP, AMC, SXT,CXM, CHL, TCY | 5 | 2(6.1) | - | | | 1(33.3) | - |
| AMP, AMC, SXT, GEN, CHL, TCY | 5 | 5(15.2) | 1(7.7) | 1(12.5) | | - | - |
| AMP, AMC, GEN, CRO,SXT, CXM | 5 | | | | 1(12.5) | | |
| AMP, AMC, SXT,CXM, GEN, TCY | 5 | 1(3) | - | | | - | - |
| AMP, AMC, SXT, GEN, CHL, TCY,CXM | 6 | | | | 2(25) | | |
| AMP, AMC, CXM, SXT, TCY, CAZ,CRO | 6 | 4(12.1) | 2(15.4) | 2(25) | 2(25) | | |
| AMP, AMC, CXM, CRO, SXT, CHL, TCY | 6 | 6(18.2) | | | | 1(33.3) | - |
| AMP, AMC, CXM, CAZ**, CRO**, SXT, CHL | 5 | | 2(15.4) | 1(12.5) | | - | - |
| AMP, AMC, GEN, SXT, CHL, TCY, CXM | 6 | | | | | - | 1(50) |
| AMP, AMC, CXM, CAZ, CRO, SXT,TCY,CHL | 6 | 4(12.1) | 2(15.4) | | | - | - |
| AMP, AMC, GEN, CIP, CHL, TCY,SXT,CXM | 7 | - | 2(15.4) | | 1(12.5) | | |
| AMP, AMC, CXM, CRO, SXT, CHL,CIP, TCY | 7 | - | | 1(12.5) | | | |
| Total | | 33(100) | 13(100) | 8(100) | 8(100) | 3(100) | 2(100) |

Key

*: the same class of antibiotics

**: the same class third generation cepalosporins, AMP: Ampicillin, CRO: Ceftriaxone, CHL: Chloramphenicol PEN: Penicillin, CIP: Ciprofloxacin, TCY: Tetracycline, GEN: Gentamicin, CXM: Cefuroxime, SXT: Sulfamethoxazole/trimetoprim, AMC: Amoxacilline-clavunic acid, CAZ: Ceftazidime

India (56%) [24]. However, this was lower than study conducted in Gurage, Ethiopia (69.2%) [55], Sweden (86%) [23] and India (79%) [56]. But, this study was higher than study conducted in Saudi Arabia (30%) [41] and Bangladesh (19%) [42]. The observed differences might be the result of variations in stethoscope routine disinfecting practice.

By profession, the contamination rate of inanimate objects was higher in laboratory professionals' mobile phone, intern students' gown and nurses' stethoscope. This was in agreement with result reported from Bahir Dar medical laboratory professionals, nurses and medical intern students, respectively [9]. The highest frequency of inanimate object contamination with bacteria was observed among HCPs' mobile phone working in laboratory unit, and

HCPs' stethoscope and gown in NICU/ ICU. This was consistent with results reported from Egypt laboratory unit [13], Jimma ICU, Ethiopia [57].

Among from the total bacterial isolates, (63.4%; 147/232) and (36.6%, 85/232) were Gram-positive and Gram-negative bacteria, respectively. It was supported by previous studies at Bahir Dar (84.9%, 15.1%) [9], and Jimma (78.9%, 21.1%), respectively [45]. The reason for high prevalence of Gram-positive bacteria, direct contact of inanimate objects to human skin flora which contains mostly Gram-positive bacteria, Gram-positive bacteria alive for a long period or Gram-positive can tolerate inanimate objects for a long period of time, but Gram-negative bacteria have short lifespan.

In the present study, *S. aureus* (22.1%, 95% CI: 17.9–26.6%), *S. epidermidis* (17.0%, 95%CI: 13.4–21.2%), *E. coli* (8.8%, 95%CI: 6.1–12.1) and *P. aeruginosa* (4.9%; 95%CI: 2.86–7.46) were the most common isolates. The majority of earlier research, conducted in Ethiopia [37, 45] and outside the country, in Nigeria and Zagazig [38, 51] reported comparable bacterial isolates with varying rates of isolates. Similarly, in this study, *S. aureus* isolates were lower than studies conducted in Asella, Ethiopia (45%) [58], and Karachi (70%) [10]. *S. aureus* is an opportunistic pathogen that frequently colonizes human skin as well as respiratory tract and might be transmitted to HCPs' inanimate objects via sneezing and coughing. However, if it gets passed the skin's protective layer, it can cause a range of acute and chronic infections, pyogenic and systemic infections. respiratory tract and might be transmitted to the inanimate objects via coughing and sneezing.

*S. epidermidis* was the second most frequent isolates in the current study. However, the magnitude of the present study (17.0%, 95%CI:13.4–21.2%) was in line with previous studies done in Debre Berhan, Ethiopia (16.7%) [59], Harar, Ethiopia (14.4%) [37] and Benegal (14.47%) [60]. This was lower than the previous studies done in Bahir Dar (44%) [9], and Jimma (60.6%) [44]. This discrepancy might be due to differences in inanimate objects decontamination practice, hygiene practice and regular hand washing with antiseptic solutions in clinical settings.

From Gram-negative bacteria, *E. coli* (8.8%, 33/376; 95%CI: 6.1–12.1%) was the most predominant bacteria followed by *P. aeruginosa* (4.9%; 18/376; 95%CI: 2.86–7.46%). This was in line with study conducted in Gondar (6.8%) *E. coli* [45] and in Harare, Ethiopia (3.7%) [37]. However lower than result reported from Mizan-Tepi, Ethiopia (15.9%) *E. coli* [61], and (19.2%) *P. aeruginosa* [62]. But higher than reports from the previous study done Bahir Dar (0.24%) *P. aeruginosa* [9] and Benegal (1.32%) *E. coli* [60]. This discrepancy might be due to differences in inanimate objects decontamination, hygiene practice and regular hand washing with antiseptic solutions in clinical settings.

In the current study, highest risk of bacterial contaminations of inanimate objects were found in those healthcare professionals working at Gyn/Obs (AOR: 8.69; 95%CI: 1.09–69.41; $P$ = 0.041) compared to other working wards. This is comparable with previous studies conducted in Bahir Dar [9]. On contrary, highest risk of bacterial contamination was found in ICU from Debre Berhan, Ethiopia [59], at medical and surgical ward in Madda Walabu University Goba Referral Hospital, Ethiopia [63] and at the laboratory unit in Alexandria, Egypt [13]. The reasons for this discrepancy could include the following: variations in the study environment, variations in the wards' cleaning practices and insufficient adherence to infection control protocols, variations in the patient load, and variations in the frequency of HCWs' interactions with patients.

Different factors were associated with the contamination of inanimate objects in the present study. Healthcare professionals' mobile phones that did not regularly disinfected were 2.7 times more likely contaminated with bacteria compared to those who had regularly cleaned their mobile phones. This was consistence with the previous results reported from Harar,

Ethiopia [37] and Debre Berhan, Ethiopia [59]. Similarly, healthcare professionals who carry mobile phone with patient care equipments were more than two times more likely contaminated with bacteria compared to those who did not carry mobile phone with patient care equipments. This study was unsupported by studies conducted in Harar, Ethiopia [37] and Gondar, Ethiopia [45]. This difference might be due to differences in habit of mobile phone keeping.

In addition, healthcare professionals who did not regularly disinfect their stethoscopes after examining each patient were 3.1 times more likely contaminated with bacteria compared to those who had regularly cleaned their stethoscopes after examining each patient. This is in line with previous studies conducted in Medda Walabu University Goba Referral Hospital, Ethiopia [63], Eastern Ethiopia [64], Benegal [60], Nepal [65] and Bangladesh [42]. This study was contrary to a study conducted in Saudi Arabia [41]. This difference might be the habit of keeping stethoscope and study setting.

Similarly, healthcare professionals who did not take any training in infection prevention were 3.9 times more likely contaminated with bacteria compared to those who had infection prevention training. This was in agreement with a study reported from Bale-Goba, Ethiopia [63]. However, this disagrees with a report from Harar, Ethiopia [37]. Poor adherence to infection prevention practice in the current study area might be the cause of this disparity.

The most serious health risk is resistance to one or more antimicrobials [58, 66]. In this study, ampicillin was highly resistant to *E. coli* (97%; 95%CI; 82.5–99.8), *Klebsiella* spp. (92.3%; 95%CI; 62.1–99.6), and *Citrobacter* species (87.5%; 95%CI: 46.7–99.3). This was in line with previous study done in Harare (78.6), (75%) *E. coli* and *Citrobacter* species, respectively [37] and Mekelle (90.9) [67].

Trimethoprim-sulfamethoxazole was highly resistant to *E. coli* (93.9%; 95%CI: 78.5–98.8), *Klebsiella* species (92.3%; 95%CI; 62.1–99.6) and *Citrobacter* species (87.5%; 95% CI:46.7-.99.3). This was consistent with previous studies done in Nigeria (90%,) *Klebsiella* species and (83.3%) *E. coli* [39], Harar (62.5%) *Citrobacter* species [37]. However, meropenem (0%) and ciprofloxacin (6.1%), for *E. coli*, and (0%) and (15.4%) for *Klebsiella* species showed lower resistance rates, respectively which agreed with earlier researches conducted in Gondar 0% [68] and Harar 22.2% [37], respectively.

In the current study *S. aureus* isolates showed high level of resistance to penicillin (97.6%; 95%CI: 82.5–99.9). This was consistent with previous studies reported from Jimma, Ethiopia (88.2) [44] and Bahir Dar, Ethiopia (82.7%) [9]. However, it was relatively lower than a study in Zambia (100%) [69]. But it was higher compared to Harar (61.3%) [37]. This might be due to differences in consumption of penicillin in the study area and they might have been common contamination sources by patients, the hospital environments, or some other unidentified sources, frequently used and overused of penicillin. While this study also showed highly sensitive to clindamycin (91.6%; 95%CI: 79.3–96.4)) and gentamycin (84.3%; 95%CI: 78.9–99.9). This was in line with studies conducted in Ethiopia, Bahir Dar gentamycin (83%) [9], Jimma clindamycin (90.3%) and gentamycin (89.6%) [57], and in Nepal gentamycin (80.5) [65].

*S. epidermidis* was highly resistance to penicillin (98.4%; 95%CI: 82.9–99.9). This was in line with results reported from Turkey (88%) [70]. However, it was higher than previous study reports from Mizan (68.8%) [61], Bahir Dar (82.7%) [9] and India (79.9%) [71]. This might be due to differences in the consumption of penicillin in the study area, empirical treatment protocols, use of antibacterial as a prophylactic, easily accessible non-prescription drugs (self-medication) and drug dosage.

The prevalence of methicillin-resistant *S. aureus* was found to be 36.1% (95%CI: 25.9–47.7) based on cefoxitin resistance. This was in line with the previous study conducted in Jimma

(26.6%) [57]. However, it was lower than studies conducted in Mizan-Tepi, Ethiopia (73.3%) [61], Pakistan (88.5%) [40], another study in Pakistan (58.4%) [72], and Zambia (48%) [69]. Similarly, this is higher compared to studies in Arba Minch, Ethiopia (19.1%) [73], India (3%) [71], and another study in India (6.4%) [74]. The observed variation in antimicrobial resistance when compared to previous studies could be attributed to various factors such as different bacterial strains, hospital environments, empirical treatment protocols, the use of antibiotics as preventive therapy, self-medication, drug dosage, and prolonged administration of common antibiotics [75].

*P. aeruginosa* showed higher resistance rate against piperacillin (94.4%; 95%CI: 70.6–99.9%), ceftazidime (72.2%; 95%CI: 46.9–89.3%) and ciprofloxacin (66.6%; 95%CI: 41.2–85.6%) and lower resistance rate to meropenem (11.1%). This was in line with studies reported from Gondar 92.3% piperacillin [68], India (71.3%) ceftazidime [40], Harar 62.5% ciprofloxacin [37], Bengal (60%) ciprofloxacin [60], India (68.4%) ciprofloxacin [71] and (0%) meropenem [70]. However, it was higher compared with studies conducted in India (0%) piperacillin [71] and Hawasa (13%) ceftazidime [76]. But, it was lower than reports from Nigeria (100%) ciprofloxacin [77] and Gondar (100%) cefepime and (92.3%) ciprofloxacin [68]. The observed variation in antimicrobial resistance compared to previous studies could be attributed to various factors such as different bacterial strains, hospital environments, empirical treatment protocols, the use of antibiotics as preventive therapy, the accessibility of certain drugs without a prescription, and prolonged administration of common antibiotics [75].

The overuse and unreasonable prescription of antibiotics may give rise to bacterial strains that are resistant to many drugs [37]. The overall MDR prevalence of the bacterial isolates from inanimate objects used by healthcare professionals in the current study was (69%; 95% CI: 62.1–74.5).This was in line with the previous studies conducted in Arba Minch, Ethiopia (57.7%) [73] and in Harar, Ethiopia (69.9%) [37]. On the other hand, this finding was higher compared to studies in Debre Berhan, Ethiopia (42.9%) [59] and in Mizan-Tepi, Ethiopia (48.9%) [61]. On contrary, it was lower than the study conducted in Bahir Dar, Ethiopia (88.3%) [9]. The observed variation in antibiotic resistance may be caused by various factors such as distinct bacterial strains, hospital environments, empirical treatment protocols, use of antibacterial as a prophylactic, easily accessible nonprescription drugs (self-medication), drug dosage, and indiscriminate or prolonged use of common antibiotics [57, 60].

In this study, multidrug resistance rate among Gram-negative and Gram-positive bacteria isolates were (75.3%; 95%CI: 63.5–83%) and (65.8%; 95%CI:57.0–72.9%), respectively. This disagree with study report from Bahir Dar, Ethiopia (88.3% and 94.7%), respectively [9]. Among Gram-positive isolates, *S. epidermidis* (67.2%; 95%CI: 54.3–78.4%) and *S. aureus* (63.9%; 95%CI: 52.6–74.1%) were showed multi-drug resistance. This is in line with results reported from Harar, Ethiopia (71%, 66.1%) [37] and Debre Berhan, Ethiopia (53.8%, 64.3%), respectively [59]. However, this disagree with result reported from Bahir Dar, Ethiopia (86.4%, 88.9%), respectively [9]. This discrepancy might be due to differences in the isolation rate, inappropriate administration of antimicrobial drugs, and self-medication practice, in the study area the majority of antibiotic classes were utilized as alternative forms of treatment and might be cross contamination between patients and HCPs' inanimate objects [57].

From Gram-negative isolates *E. coli* (84.8%; 95%CI: 61.9–94.6%), *Klebsiella* species (76.9%; 95%CI: 42.5–92%), and *P. aeruginosa* (72.2; 95%CI; 35.7–89.5) were most predominantly showed MDR. This is comparable with reports from Harar, Ethiopia (86.7%) *Klebsiella* species and (87.5%) *P. aeruginosa*, respectively [37] and Bahir Dar Ethiopia (92.6%) *E. coli* [9]. However, this was lower than result reported from Bahir Dar, Ethiopia (100%) *Klebsiella* species and (100%) *P. aeruginosa* [9]. But it was lower than result reported from Mizan-Tepi (28.6%), (53.8%) and (30%), respectively [61]. This discrepancy might be due to differences in the

isolation rate, inappropriate administration of antimicrobial drugs, and self-medication prac-
tice, in the study area the majority of antibiotic classes were utilized as alternative forms of
treatment [57].

## Conclusion

The results of the present study indicated that many of the inanimate objects of HCPs were
contaminated with different bacteria. The most common bacterial isolates were *S. aureus*, *S.
epidermidis*, *E. coli* and *P. aeruginosa*. Gram-positive isolates showed high level of resistance to
penicillin and the majority of Gram-negative bacteria were resistant to ampicillin, trimetho-
prim/sulfamethoxazole, and amoxicillin-clavulanic. On the other hand, meropenem for
Gram-negative and clindamycin for Gram-positive bacteria showed higher level of sensitivity.
It also revealed that did not regularly disinfecting inanimate objects and those inanimate
objects taken from gynecology/obstetrics ward were most likely to be contaminated. More-
over, those healthcare professionals who carried mobile phone with patient care materials
were more likely to be contaminated with bacteria compared to those who did not carry
mobile phone with patient care materials. Furthermore, those HCPs who did not take infec-
tion prevention training were more likely to be contaminated.

## Recommendations

We strongly recommend to all healthcare professionals regular disinfection of inanimate
objects to minimize bacterial colonization of inanimate objects and potential spread of infec-
tion, especially drug-resistant strains. Antimicrobial treatment should be based on the result of
culture and sensitivity Also, healthcare professionals in Debre Markos Comprehensive Special-
ized Hospital should focus on not carrying mobile phone medical equipments and hospital
administrators should provide infection prevention training for all healthcare professionals.

## Limitations of the study

This study did not differentiate the extended beta-lactam spectrum, because of shortage of
drugs. Additionally, this study excluded other important bacterial pathogens that cause HAI,
such as anaerobes, due to a lack of laboratory facilities.

## Supporting information

**S1 Table. Multidrug resistance pattern of *Pseudomonas aeruginosa* isolates from inanimate
objects used by healthcare professionals, at DMCSH, Northwest Ethiopia 2023.**
(DOCX)

**S2 Table. Multidrug resistance pattern of Gram-positive isolates from inanimate objects
used by healthcare professionals at DMCSH, Northwest Ethiopia 2023.**
(DOCX)

**S1 Data. Supporting information file.**
(XLSX)

## Acknowledgments

We would like to sincerely thank Debre Markos Comprehensive Specialized Hospital
(DMCSH) administration and staffs for allowing me to conduct the research in the hospital.
Our heartfelt gratitude and special thank goes to DMCSH microbiology laboratory team for
providing materials and assisting the isolation and identification of the microorganisms. We

acknowledge the study participants for their participation without them the research would not be a reality.

## Author Contributions

**Conceptualization:** Senedu Kindie, Getachew Mengistu, Tebelay Dilnessa.

**Data curation:** Senedu Kindie, Getachew Mengistu, Mulatu Kassahun, Abebaw Admasu, Tebelay Dilnessa.

**Formal analysis:** Senedu Kindie, Getachew Mengistu, Mulatu Kassahun, Abebaw Admasu, Tebelay Dilnessa.

**Funding acquisition:** Senedu Kindie.

**Investigation:** Senedu Kindie.

**Methodology:** Senedu Kindie, Getachew Mengistu, Tebelay Dilnessa.

**Project administration:** Senedu Kindie, Getachew Mengistu, Tebelay Dilnessa.

**Resources:** Senedu Kindie.

**Software:** Senedu Kindie, Mulatu Kassahun.

**Supervision:** Getachew Mengistu, Tebelay Dilnessa.

**Validation:** Senedu Kindie, Getachew Mengistu, Mulatu Kassahun, Abebaw Admasu, Tebelay Dilnessa.

**Visualization:** Senedu Kindie, Getachew Mengistu, Abebaw Admasu, Tebelay Dilnessa.

**Writing – original draft:** Senedu Kindie.

**Writing – review & editing:** Getachew Mengistu, Mulatu Kassahun, Abebaw Admasu, Tebelay Dilnessa.

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
