## [Decision Letter · Decision Letter 0]

4 Oct 2024

PONE-D-24-37007Bacterial profile and antimicrobial susceptibility patterns of isolates from inanimate objects used by healthcare professionals and associated factors at Debre Markos Comprehensive Specialized Hospital, Northwest EthiopiaPLOS ONE

Dear Dr. Dilnessa ,

Thank you for submitting your manuscript to PLOS ONE. After careful consideration, we feel that it has merit but does not fully meet PLOS ONE’s publication criteria as it currently stands. Therefore, we invite you to submit a revised version of the manuscript that addresses the points raised during the review process.

We look forward to receiving your revised manuscript.

Kind regards,

Mengistu Hailemariam Zenebe, PhD

Academic Editor

PLOS ONE

Journal Requirements:

2. We suggest you thoroughly copyedit your manuscript for language usage, spelling, and grammar. If you do not know anyone who can help you do this, you may wish to consider employing a professional scientific editing service. The American Journal Experts (AJE) (https://www.aje.com/) is one such service that has extensive experience helping authors meet PLOS guidelines and can provide language editing, translation, manuscript formatting, and figure formatting to ensure your manuscript meets our submission guidelines. Please note that having the manuscript copyedited by AJE or any other editing services does not guarantee selection for peer review or acceptance for publication. Upon resubmission, please provide the following: ● The name of the colleague or the details of the professional service that edited your manuscript ● A copy of your manuscript showing your changes by either highlighting them or using track changes (uploaded as a *supporting information* file) ● A clean copy of the edited manuscript (uploaded as the new *manuscript* file)

3. We note that your Data Availability Statement is currently as follows: “All relevant data are within the manuscript and in Supporting Information files.”

Please confirm at this time whether or not your submission contains all raw data required to replicate the results of your study. Authors must share the “minimal data set” for their submission. PLOS defines the minimal data set to consist of the data required to replicate all study findings reported in the article, as well as related metadata and methods (https://journals.plos.org/plosone/s/data-availability#loc-minimal-data-set-definition). For example, authors should submit the following data: - The values behind the means, standard deviations and other measures reported; - The values used to build graphs; - The points extracted from images for analysis. Authors do not need to submit their entire data set if only a portion of the data was used in the reported study. If your submission does not contain these data, please either upload them as Supporting Information files or deposit them to a stable, public repository and provide us with the relevant URLs, DOIs, or accession numbers. For a list of recommended repositories, please see https://journals.plos.org/plosone/s/recommended-repositories. If there are ethical or legal restrictions on sharing a de-identified data set, please explain them in detail (e.g., data contain potentially sensitive information, data are owned by a third-party organization, etc.) and who has imposed them (e.g., an ethics committee). Please also provide contact information for a data access committee, ethics committee, or other institutional body to which data requests may be sent. If data are owned by a third party, please indicate how others may request data access.

Reviewers' comments:

Reviewer's Responses to Questions

**Comments to the Author**

1. Is the manuscript technically sound, and do the data support the conclusions?

Reviewer #1: Partly

Reviewer #2: Partly

2. Has the statistical analysis been performed appropriately and rigorously? 

Reviewer #1: Yes

Reviewer #2: No

3. Have the authors made all data underlying the findings in their manuscript fully available?

Reviewer #1: Yes

Reviewer #2: Yes

4. Is the manuscript presented in an intelligible fashion and written in standard English?

Reviewer #1: Yes

Reviewer #2: No

5. Review Comments to the Author

Reviewer #1: Comments to the Author

Abstract: didn’t show the reader why this research (no clear gap/s why this research is conducted).

In the findings reporting the point prevalence need to presented it with CI, at 95%. For example the overall prevalence was 60 %( 95%CI:_____, _____).

All pathogen names (Staphylococcus aureus) should be italized. The recommendations drawn from the conclusion are not feasible.

Introduction/background

Is there any national/WHO guideline for the disinfection of the intimate objective you have selected? If there is no guideline, it is difficult to assess the problems and put recommendations at the end of the study. For example, gowns are stated to transmit pathogen from one person to another. Do we have strategies to avoid/minimize this problems?

In the gap: it was stated that there is no report about the role of these objects (you have selected) in the study area. If similar study done elsewhere, do we expect different finding in your study site? If not, this couldn’t be a gap.

Methods

Why only three objects selected (any references for this?). Some of the independent variables are not clear (eg. Carry mobile phone with patient care materials, infection prevention manual (use/presence?), laundry gown, share stethoscopes …).

Sample size were calculated for objects. It is not clear how distributed for health care worker, wards? Why survey for stethoscopes?

There is no exclusion criteria for the study subjects. Eg. How new gown or washed can be a study subject…….

Ethical issue. Nothing was Saied about the health care provider using objects contaminated with bacterial pathogens/drug resistant pathogen. What should be done for these group?

Results

In the presentation of age participants, you use mean. But it should be reported with SD (+5.5). it also not important to present the detail of age( you may refer to table).

Is 25% reports as “most?”

In assess the practice of health care provider in using these selected object, what approach was used? Is it observation? Interview?

It is also report 71.4% as more than 3/4th

To indicate the difference in the level of contamination between different wards, objects it is must to show the difference is significantly different (use P-value).

Regular disinfection of mobile phone (AOR: 2.69; 95% CI: 1.15-6.25; P=0.021),

Disinfection of stethoscope after examining each patient (AOR: 3.06, 95% CI: 1.23-7.59; P=0.016) were associated with bacterial contamination of HCPs inanimate objects. These finding indicate disinfection is not important and is risk for contamination. These report is different from what were presented in the table 5. Please rephrase these.

The major pathogen indicated in the result was Staphylococcus aureus. But there is no report regarding Methicillin-resistant Staphylococcus aureus, the leading causes of hospital-acquired infections

Discussion

It is too broad compared to the major findings. Need to revise. The discussion should focus on major findings. What is the clinical implication of your findings? What modifiable variable to be recommended.

The conclusion is not focused. Please conclude the finding based on specific objectives.

Reviewer #2: Please revise the write up/flow of the manuscript parts of Title ,abstract, introduction, methology, result, Discussion prats as based the recommended comments and the flow of the language parts of the manuscript

6. PLOS authors have the option to publish the peer review history of their article (what does this mean?). If published, this will include your full peer review and any attached files.

Reviewer #1: **Yes: **Kedir Urgesa Bofe

Reviewer #2: No

---

## [Author Response · Author response to Decision Letter 0]

13 Oct 2024

Dear Dr. Mengistu Hailemariam Zenebe

Academic Editor, PLoS One

This is a revised manuscript (PONE-D-24-37007) entitled: “Bacterial profile and antimicrobial susceptibility patterns of isolates from inanimate objects used by healthcare professionals at Debre Markos Comprehensive Specialized Hospital, Northwest Ethiopia” to be considered for publication as research article in “Plos one”. 

By: Senedu Kindie, Getachew Mengistu, Mulatu Kassahun, Abebaw Admasu, Tebelay Dilnessa*, 

First, we would to acknowledge you for the decision and comments forwarded which is very important for the improvement of the paper. We would like to thank very much reviewers for providing important comments and questions for the improvement of our manuscript. Please find the attached revised documents that were accommodated with the comments forwarded by editor and reviewers. We gave a point-by-point response to each comment below. The comments were accepted and incorporated to the paper. We also gave brief explanations for the concerns raised by the editor and reviewers. Additionally, a proofreading was made thoroughly to improve the manuscript. 

Responses to Academic Editor:

1. Editor: Please ensure that your manuscript meets PLOS ONE's style requirements, including those for file naming.

Author response: Thank you for sharing the PLOS ONE's style requirements and templates. We formatted the manuscript based on authors guideline in preparation of our manuscript.

2. Editor: We suggest you thoroughly copyedit your manuscript for language usage, spelling, and grammar. 

Author response: We revised through rephrasing, editing, correcting and proofreading all the contents of the manuscript. 

3. Editor: Please review your reference list to ensure that it is complete and correct. If you have cited papers that have been retracted, please include the rationale for doing so in the manuscript text, or remove these references and replace them with relevant current references. Any changes to the reference list should be mentioned in the rebuttal letter that accompanies your revised manuscript. If you need to cite a retracted article, indicate the article’s retracted status in the references list and also include a citation and full reference for the retraction notice.

Author response: Thank you for sharing the PLOS ONE's journal requirements for references. We went through each reference and checked online on databases its availability. We got all references used in the manuscript were complete, correct and available in the databases. There are no retracted references used in this manuscript. 

Responses to Reviewer #1

Thank you for your critical observations and forwarded comments and questions to us for the betterment of our manuscript. We accepted the comments and incorporated to the manuscript and we gave some responses point by point to your comments. 

• Title: Reviewer 1: Revise the title as ‘‘Prevalence and antimicrobial susceptibility of bacteria on healthcare professionals’ equipment in Debre Markos Hospital’’

Author response: Thank you for the comment. We revised the title in a certain extent, but we prefer ‘bacterial profile’ than ‘prevalence’ because we expect the phrase ‘bacterial profile’ more explanatory for the question; What are the bacteria present in HCPs’ mobile and medical equipment’s. Additionally, we prefer ‘inanimate objects’ to refer mobile phone, statoscope and gown. Here, we expected that gown was not considered as equipment.

Abstract

1. Reviewer 1: Abstract didn’t show the reader why this research (no clear gap/s why this research is conducted). 

Author response: Thank you for the comment. I revised the abstract based on the recommendation.

2. Reviewer 1: In the findings reporting the point prevalence need to presented it with CI, at 95%. For example the overall prevalence was 60 % (95%CI:_,_). All pathogen names (S. aureus) should be italized. The recommendations drawn from the conclusion are not feasible.

Author response: Thank you, I revised based on the comments forwarded. Please see the abstract part. See the revision line 52. Similarly, we italized the name of the microorganisms throughout the manuscript.

Introduction

3. Reviewer 1: Is there any national/WHO guideline for the disinfection of the intimate objective you have selected? If there is no guideline, it is difficult to assess the problems and put recommendations at the end of the study. For example, gowns are stated to transmit pathogen from one person to another. Do we have strategies to avoid/minimize these problems? In the gap: it was stated that there is no report about the role of these objects (you have selected) in the study area. If similar study done elsewhere, do we expect different finding in your study site? If not, this couldn’t be a gap.

Author response: Thank you for the concern. Of course, there are infection prevention guidelines in WHO and various countries, but there was no infection prevention guideline in Ethiopia for this specific area. I have already compared and contrasted the previous researches to our research, clearly there are variations and similarities between them based on the different parameters.

In this area, there were researches worldwide, but no/less research was conducted in Ethiopia and no specifically to our study area. Through publication, I hope the Ministry of Health of Ethiopia will find it and may start to develop guideline, initiated from our conclusion and recommendation. 

Methods

4. Reviewer 1: Why only three objects selected (any references for this?). Some of the independent variables are not clear (e.g. Carry mobile phone with patient care materials, infection prevention manual (use/presence?), laundry gown, share stethoscopes …). 

Author response: We selected these three objectives because each of this equipment is routinely used on daily bases. Therefore, the contribution of this three equipment for the transmission of infectious agent is more significant and that is why we selected. In the case of carry mobile phone with patient care materials is to mean that ‘is the healthcare professional care mobile with gauze, blade, tourniquet, syringe, etc. In the cause of infection prevention manual, some departments have, and others not. It is expected that those who have the infection prevention manual likely to follow the infection prevention protocol. Laundry gown means that ‘Do you wash your gown regularly?’ Some times in the case of shortages of statoscopes, physicians may share statoscopes and we want to see it and sharing also may contribute for the contamination of gowns. 

5. Reviewer 1: Sample size was calculated for objects. It is not clear how distributed for health care worker, wards? Why survey for stethoscopes? 

Author response: The sample size was calculated based on the sample size calculation principle to be representative to the three objects. The sample size was proportionally allocated to different healthcare professionals and working wards according to their population size. Stethoscope swabs were collected from health professionals who have stethoscopes by using a census until the required sample size is fulfilled because of the proportionally allocated samples is larger than the health professionals working in the hospital.

6. What does it mean ‘all inanimate objects?

Author response: We revised it accordingly. We operationalized the phrase ‘inanimate objects’ as ‘Inanimate objects: are gown, mobile phones and stethoscopes used by healthcare professionals at DMCSH.

7. Reviewer 1: There were no exclusion criteria for the study subjects. E.g., How new gown or washed can be a study subject……. 

Author response: During the study period, there were no allocation of gowns. Therefore, we are sure and we would not include to the exclusion criteria. But ‘gown washing habit’ was assessed and it was one of our objectives to be addressed as a risk factor for transmission of infection.

8. Reviewer 1: How many samples were taken from each profession, and are all equipment samples were taken from each individual?

Author response: From a total of 191 study participants, 376 swabs were taken (the detail of each inanimate objects and health professional was found in Figure 1)

9. Reviewer 1: Ethical issue. Nothing was said about the health care provider using objects contaminated with bacterial pathogens/drug resistant pathogen. What should be done for these group?

Author response: Because our study subjects were inanimate objects (gown, mobile phone and stethoscope), they serve as a carrier, but they told nothing about the status of the healthcare professionals’ status. Therefore, there was no ethical issue in this respect. But the healthcare professionals will serve from the recommendations. 

Results

10. Reviewer 1: In the presentation of age participants, you use mean. But it should be reported with SD (+5.5). it also not important to present the detail of age (you may refer to table). Is 25% reports as “most?” 

Author response: Thank you revisions were made based on the comments. For the age group, we grouped based on the fork force in which the profession starter (20 years age) with five years interval to the more experienced professionals (above 35 years old). There are seven (7) field of specialization based on our categorization, therefor 25.7% can be majority based on these seven groups.

11. Reviewer 1: In assess the practice of health care provider in using these selected object, what approach was used? Is it observation? Interview?

Author response: Both approaches (interview and observation) were used to collect the questioner-based data. But the former was predominantly used.

12. Reviewer 1: It is also report 71.4% as more than 3/4th, to indicate the difference in the level of contamination between different wards, objects it is must to show the difference is significantly different (use P-value).

Author response: We revised based on the recommendations forwarded by the reviewer.

13. Reviewer 1: Regular disinfection of mobile phone (AOR: 2.69; 95%CI: 1.15-6.25; P=0.021),

Disinfection of stethoscope after examining each patient (AOR: 3.06, 95% CI: 1.23-7.59; P=0.016) were associated with bacterial contamination of HCPs inanimate objects. These finding indicate disinfection is not important and is risk for contamination. These reports are different from what were presented in the table 5. Please rephrase these.

Author response: Thank you for the critical comment. We corrected it accordingly. See page 17, lines 360-366 and Table 5.

14. Reviewer 1: For what purpose chocolate agar was used?

Author response: Chocolate was intended to isolate fastidious bacteria such as S. pneumoniae, H. influenzae, etc.

15. Reviewer 1: The antimicrobials listed were unmatched with table in the result

Author response: Thank you for the observation. We corrected and made matched the list of antimicrobials with the result.

16. Reviewer 1: In the prevalence part, the number (376) is number of personnel or sample?

Author response: The number of HCPs were 191 from which 376 swabs were collected. The details of each professional were indicated in Table 1.

17. Reviewer 1: Please use the correct standard MDR classification and recalculate (AMP and AMC are the same class)

Author response: We revised the texts and tables based on the definition of MDR and categories of antibiotics.

18. Reviewer 1: The major pathogen indicated in the result was S. aureus. But there is no report regarding Methicillin-resistant S. aureus, the leading causes of hospital-acquired infections

Author response: There was a statement regarding MRSA, and now we added more on this regard. See the page 22, lines 409-417.

Discussion

19. Reviewer 1: It is too broad compared to the major findings. Need to revise. The discussion should focus on major findings. 

Author response: Thank you. We revised the conclusion based on our major findings.

Conclusion

20. Reviewer 1: What is the clinical implication of your findings? What modifiable variable to be recommended. The conclusion is not focused. Please conclude the finding based on specific objectives.

Author response: We revised the conclusions and recommendations based on our objectives.

21. Revise the introduction, method, conclusion and recommendation, etc.

Author response: Thank you for the comment. We revised all portions which was requested by the reviewer.

Responses to Reviewer #2

Thank you for your critical comments and questions to us for improvement of our paper. We accepted comments and incorporated to the paper and we gave some responses point by point to your comments.

1. Reviewer 2: Please revise the write up/flow of the manuscript parts of title, abstract, introduction, methodology, result, discussion parts as based the recommended comments and the flow of the language parts of the manuscript

 Author response: Thank you for the comments and we revised in each part of the manuscript. See the attached manuscript and manuscript with track changes.

Finally, once again we would like to thank you.

In case of any questions and doubts, please do not hesitate to contact us anytime.

 With best regards,

 Tebelay Dilnessa 

 Onbehaf of all authors

---

## [Editor Report · Decision Letter 1]

25 Oct 2024

Bacterial profile and antimicrobial susceptibility patterns of isolates from inanimate objects used by healthcare professionals at Debre Markos Comprehensive Specialized Hospital, Northwest Ethiopia

PONE-D-24-37007R1

Dear Dr. Dilnessa ,

We’re pleased to inform you that your manuscript has been judged scientifically suitable for publication and will be formally accepted for publication once it meets all outstanding technical requirements.

Kind regards,

Mengistu Hailemariam Zenebe, PhD

Academic Editor

PLOS ONE
---

## [Editor Report · Acceptance letter]

30 Oct 2024

PONE-D-24-37007R1 

PLOS ONE

Dear Dr. Dilnessa , 

I'm pleased to inform you that your manuscript has been deemed suitable for publication in PLOS ONE. Congratulations! Your manuscript is now being handed over to our production team.

Kind regards, 

on behalf of

Dr. Mengistu Hailemariam Zenebe 

Academic Editor

PLOS ONE